SUTrans-NET: a hybrid transformer approach to skin lesion segmentation

Li Yaqin
Tian Tonghe
Hu Jing
Yuan Cao yc@whpu.edu.cn
School of Mathematics and Computer Science, Wuhan Polytechnic University School , Wuhan , Hubei , China
Coelho Paulo
Electronic publication date: 2024 Mar 13
Publication date: 2024
Volume: 10
Electronic Location ID: e1935
Received 2023 Sep 15; Accepted 2024 Feb 18
Copyright: ©2024 Li et al.
Copyright year: 2024
Copyright holder: Li et al.
License: This is an open access article distributed under the terms of the Creative Commons Attribution License, which permits unrestricted use, distribution, reproduction and adaptation in any medium and for any purpose provided that it is properly attributed. For attribution, the original author(s), title, publication source (PeerJ Computer Science) and either DOI or URL of the article must be cited.
License URL: https://creativecommons.org/licenses/by/4.0/

Keywords: Skin lesion segmentation, Transformer, Edge details, Image feature extraction

Funding: The authors received no funding for this work.

==============================
Melanoma is a malignant skin tumor that threatens human life and health. Early detection is essential for effective treatment. However, the low contrast between melanoma lesions and normal skin and the irregularity in size and shape make skin lesions difficult to detect with the naked eye in the early stages, making the task of skin lesion segmentation challenging. Traditional encoder-decoder built with U-shaped networks using convolutional neural network (CNN) networks have limitations in establishing long-term dependencies and global contextual connections, while the Transformer architecture is limited in its application to small medical datasets. To address these issues, we propose a new skin lesion segmentation network, SUTrans-NET, which combines CNN and Transformer in a parallel fashion to form a dual encoder, where both CNN and Transformer branches perform dynamic interactive fusion of image information in each layer. At the same time, we introduce our designed multi-grouping module SpatialGroupAttention (SGA) to complement the spatial and texture information of the Transformer branch, and utilize the Focus idea of YOLOV5 to construct the Patch Embedding module in the Transformer to prevent the loss of pixel accuracy. In addition, we design a decoder with full-scale information fusion capability to fully fuse shallow and deep features at different stages of the encoder. The effectiveness of our method is demonstrated on the ISIC 2016, ISIC 2017, ISIC 2018 and PH2 datasets and its advantages over existing methods are verified.

Introduction

Skin cancer can be divided into three types: basal cell carcinoma, squamous cell carcinoma, and melanoma, of which melanoma is the most deadly. In its early stages, skin cancer is difficult for patients to detect with the naked eye, and the characteristics of the lesions are very similar to those of benign skin conditions such as psoriasis and eczema, which usually require computer-aided diagnostic (CAD) tools to aid in diagnosis. Skin cancer is one of the fastest growing cancers in the world, with survival rates as high as 90% if diagnosed and treated early. However, due to the low contrast of the lesion to normal skin, patients often do not detect skin cancer until it is in the mid to late stages. Therefore, early detection and treatment remains one of the most important ways to improve the survival rate of skin cancer patients. Automated segmentation of skin lesions can effectively improve the accuracy of melanoma diagnosis. It can effectively segment lesions on patients’ skin in the early stages of the lesion, further preventing melanoma from progressing to melanoma.

To effectively detect skin lesion tissue, dermatologists typically use dermoscopy to visualize lesions and surrounding tissues. However, the subjective judgment of experts and the differences in skin characteristics of different patients can affect the effectiveness of manual segmentation. To solve this problem, Ashour et al. (2018) proposed a neutral C-mean clustering (NCM) method based on the histogram clustering estimation (HBCE) algorithm to determine the number of pixel clusters in dermoscopy images. Conventional machine learning algorithms can also be used for image classification of skin lesions. However, since deep convolutional neural networks (CNNs) (Krizhevsky, Sutskever & Hinton, 2012; Simonyan & Zisserman, 2014; Szegedy et al., 2015; Lo et al., 1995) are used for more downstream tasks, deep learning for skin lesion segmentation is more accurate than traditional segmentation methods. It can have better generalization for different cases of skin lesions.

In recent years, deep learning for medical image segmentation has shown more potential than traditional medical image segmentation methods. Such as thresholding, edge detection, and horizontal integration methods, Dar & Padha (2019) they are usually based on some heuristic rules or specific mathematical operations to segment images into different regions, however, these methods perform poorly when dealing with complex scenes and images with diversity. The proposal of fully convolutional neural network (FCN) (Long, Shelhamer & Darrell, 2015) has made a breakthrough in medical image segmentation. High-Resolution Network (HRNet) (Sun et al., 2019) segmented the network by multi-scale fusion and fused the low-resolution features into high-resolution features for better extraction of image features. Radman, Sallam & Suandi (2022) proposed Deep Residual Network (ResNet). The proposal of a residual network further solves the problem of degradation of deep networks. The network can extract deeper image features, which lays a historical foundation for training the network to a deeper level. U-Net (Ronneberger, Fischer & Brox, 2015) is a network based on an encoder–decoder architecture, which combines high semantic information extracted by the encoder with the original image resolution reduced by the decoder through hopping connections and is used for medical image segmentation. Milletari, Navab & Ahmadi (2016) proposed a V-Net network for three-dimensional (3D) medical image segmentation, which can achieve clinical organ segmentation quickly and accurately by optimizing the objective function based on the Dice overlap coefficient. Zhou et al. (2019) proposed a U-Net++ network for medical image segmentation, which continuously performs upsampling and splicing operations on the previous layer to avoid losing the semantic information of the image, and it is easier to perform pruning operations at different network depths. Peng et al. (2021) proposed Local Context Perception Net (LCP-Net), which can achieve clinical organ segmentation quickly and accurately by proposing Parallel Dilated Convolution (PDC) and Local Context Embedding (LCE) to obtain rich feature graph information. The “U”-shaped network architecture is widely used in medical segmentation due to its unique advantages, effectively segmenting small medical datasets. However, most of the existing CNN methods cannot establish long-term dependencies and global contextual links due to the limitation of sensory fields in the convolutional operation. The repetitive step and pooling operations inevitably lose the image’s resolution, and the loss of positional and full-text contextual information affects the final segmentation accuracy.

The application of deep learning to medical image segmentation has developed rapidly. Classical network structures such as FCN, High Resolution Network (HRNet), Deep Residual Network (ResNet), and U-Net are widely used in medical image segmentation tasks, but these methods have limitations in dealing with global contextual information and long-term dependencies. With the successful application of Vision Transformer (ViT) (Dosovitskiy et al., 2020) in natural language, Transformer is gradually making inroads into the image domain, and various Transformer network architectures are being used for more downstream tasks (Touvron et al., 2021; Liu et al., 2021; Chang et al., 2021) that are capable of encoding long-term dependencies. Chen et al. (2021b) propose TransUNet (Transformers and U-Net), which uses Transformer to encode labeled image patches and then directly upsamples the hidden feature representations, and the emergence of this model opens up the application of Transformer in the field of medical image segmentation. However, although Transformer excels in global context modeling, it lacks the spatial information of the image and has limitations, especially in capturing the image structure boundaries. For this reason, some methods that combine the local information processing capability of the “U” network and the powerful global information modeling capability of the Transformer have recently emerged, such as U-Transformer (Petit et al., 2021), FATNet (Wu et al., 2022), Attention-UNet (Oktay et al., 2018), and so on. These methods take advantage of the Transformer’s long-term dependence on modeling to further optimize the CNN extracted features. The CNN extracted features are further optimized for medical image segmentation tasks. These methods have achieved some success in medical image segmentation tasks. Although these methods have great potential, their application to medical image datasets with small amounts of data is still challenging.

Although there have been some medical image segmentation methods combining CNN and Transformer that perform well in practice, most of these methods currently use the existing backbone (e.g., ViT, ResNet) directly to extract information, and there is a lack of information flow between layers, so the segmentation performance improvement is not apparent. In addition, the multi-head self-attention mechanism in Transformer is computationally intensive and is not applicable to small datasets such as medical images but is more suitable for text and large datasets. In medical image segmentation, the segmentation of lesion boundaries is very critical. It requires the network to learn the long-term dependencies between pixels thoroughly, both the spatial information of the image and the global context modeling to complement the spatial and textural information of the image. Therefore, a more appropriate Transformer structure for medical images must be designed to improve segmentation performance and reduce computational complexity.

In order to solve the above problems, we propose a new encoder–decoder architecture called SUTrans-NET. Overall, we construct a medical image segmentation model that fully combines the advantages of CNN and Transformer. In SUTrans-NET, CNN and Transformer form a dual decoder in parallel to extract spatial features required for image segmentation by utilizing the CNN branch’s advantage in mining shallow texture information. The Transformer branch encodes the input image at the same time to efficiently capture global contextual information in the medical image. The constructed dual-encoder network carries out each stage of the information interaction to realize the sharing of local and global information. Inspired by the Spatial group-wise Enhancement (SGE) module (Li, Hu & Yang, 2019), we propose a spatial attention SpatialGroupAttention (SGA) module, which improves the method of group channel attention. Meanwhile, we replace the traditional Embedding layer with the Focus module (Zhu et al., 2021) in Yolov5 to allow the Transformer branch to learn the sensory field of different levels of images while interacting with the CNN branch for information. On the other hand, the Squeeze-and-Excitation (SE) module (Hu, Shen & Sun, 2018) is introduced in the Transformer to allow the CNN and the Transformer to share the weight values by explicitly modeling the interdependence between the channels in order to allow a high degree of fusion between the local and global contexts. We design a decoder with full-scale information fusion capability to fuse shallow and deep features at different encoder stages fully. Unlike the traditional Atrous Spatial Pyramid Pooling (ASPP) (Chen et al., 2017), we do slicing operation on the input features, and slicing operation on the channels makes it easier to encode multiscale objects, thus learning the task of dense prediction. We demonstrate the effectiveness of SUTrans-NET on four datasets, ISIC 2016, ISIC 2017, ISIC 2018 and PH2, and our work is summarized as follows:

1. We propose an innovative SUTrans-NET framework to solve complex problems in medical image segmentation. Unlike traditional methods, we adopt a dual-decoder structure to build the CNN and Transformer in parallel and realize the dynamic interaction of information between them, which solves the problem of information illiquidity between layers and thus realizes the sharing of global and local information. SUTrans-NET not only effectively solves the dependencies between pixels and alleviates the problem of the pure Transformer’s poor performance on small medical image datasets but also improves CNN’s ability to establish long-term dependencies and global contextual connections.

2. In our encoder–decoder architecture, we replace the traditional Multi-head Self-Attention (MHA) module (Dosovitskiy et al., 2020) with our constructed SGA module, which utilizes the idea of grouped attention, and the SGA module can improve the learning ability of sub-features. These sub-features can represent different semantic entities, and each group can self-improve the expression of information. The constructed Spatial Attention SGA module can reduce the computational cost of the Transformer and, at the same time, provide the spatial information needed for medical image segmentation to the Transformer branch. We also introduce the SE module in SUTrans-NET, which focuses on correlations and dependencies between channels, suppresses less critical information and pays attention to attention-worthy information, and can display recalibration of different features, allowing the model to focus on more helpful image features, thus improving segmentation performance.

3. Our full-scale decoder employs an information fusion strategy at different stages and scales. It redesigns the traditional simple jump-joining approach by slicing the CNN branches to extract features with different receptive fields. This approach alleviates the problem of image resolution loss due to repetitive step and pooling operations during upsampling and provides richer local information.

Related works

Application of “U” network for medical image segmentation

In traditional methods, skin lesion segmentation usually uses histogram thresholding (Emre Celebi et al., 2013), region growing method (Ma & Tavares, 2015), clustering method (Suer, Kockara & Mete, 2011; Huang, Kang & Xu, 2020), edge and region-based method (Abbas et al., 2011), etc., which often can only extract the low-level features of the image, have a strong dependence on the threshold value, and have a high requirement on the image feature definition. With the great splendor of deep learning (Krizhevsky, Sutskever & Hinton, 2012) in computer vision, CNN-based methods are gradually applied to skin lesion segmentation and achieve good results.

U-Net (Ronneberger, Fischer & Brox, 2015) has been widely used in image segmentation due to its unique network structure. It is currently achieving good results in diagnosing viral diseases such as tumors, breast cancer, liver cancer, etc. (Alom et al., 2018) combined RNN and ResUNet to perform image segmentation and used the residual convolutional layer recursive processing to obtain the accumulation of features, further improving the representational ability of segmentation. Ma, Zou & Liu (2021) added a context module to jump connections in the MHSU-Net network to reduce feature loss during segmentation. Chen et al. (2021a) proposed cross-scale residual network (CSR-Net), which realizes the fusion of features from different layers through cross-scale residual connections and realizes the transformation between different scales and channels of the hierarchy to establish a close connection at different resolutions. Soulami et al. (2021) proposed an end-to-end improved UNet model that combines residual fast and hybrid loss functions for detecting, segmenting, and classifying breast masses in one stage.

Zhang et al. (2022) proposed dense-dual-task network (DDTNet).The feature fusion strategy proposed by DDTNet fuses contextual features with cancer cell location information, and the detection and segmentation modules share the same backbone network, which further senses the boundary information for accurate breast cancer lesion segmentation. To better utilize the global information, CMM-Net (Al-Masni & Kim, 2021), MCNet (Wang, Hu & Lyu, 2020), GC-DCNN (Lan et al., 2020), and other models improved the pyramid pooling module in UNet. Al-Masni & Kim (2021) proposed the contextual multiscale multilevel CMMNet network, which fused the traditional convolutional network with global contextual features at multiple spatial scales and designed a new dilated convolutional module so that the feature maps expand the sensory field by adjusting the rate according to their size. Wang, Hu & Lyu (2020) proposed the medical segmentation problem of a multi-path connected network (MCNet), which uses the pyramid pooling layer to retain semantic and spatial information, and integrates the multiple paths generated by its aggregation in the coding stage to obtain a larger receptive field using multiscale features. Lan et al. (2020) proposed the global context-based dilated convolutional neural network (GC-DCNN); GC-DCNN uses the pyramid pooling module to learn additional discriminative features and fuses them for more robust feature representation. Yuan et al. (2021) proposed a network ResD-Unet for pulmonary artery segmentation; in this architecture, a new dense, fast refinement of image segmentation is introduced, which also performs well on small datasets. Lee et al. (2020) proposed a network consisting of multiple U-Nets (multiscale U-Net, MU-Net), using a multiscale approach through joint convolutional inputs, capable of removing noise at different scales to learn to build images with different coarseness and fineness, with sequential output targets from low to high frequencies. Although the CNN-based network can capture the positional information of the image very well, the effect on global context-dependent processing is fragile.

Transformer

Transformer’s outstanding performance in natural language has been valued by many computer vision experts and introduced to many downstream tasks. CNN is only good at acquiring local features due to its limited sensory field.Transformer, which can acquire global features, is used to optimize the automatic segmentation technique for medical images. Transformer shines in NLP tasks, and ViT Applying Transformer to image classification tasks, TransUNet (Transformers and U-Net) model (Chen et al., 2021b) opened the application of Transformer in medical image segmentation.

 Li et al. (2020) proposed an attention-based nested U-Net (ANU-Net),which adds an attention mechanism to the convolutional blocks of ANU-Net to obtain resolution features at different levels of semantics. Tang et al. (2021) proposed DualAttention-based Dense SU-net (DA-DSUnet), where the attention module ensures relative dependence in position and better preserves the boundary information in the image, which is not easy to segment. CHANG Y proposed the three-branch network structure ClawU-NETwithTransformers, TransClaw combines CNN operations with Transformer operations to obtain global context information as well as local features captured by convolution on the deep network to achieve better detail segmentation. Zheng et al. (2021) proposed a Segmentation Transformer SEgementation TRansformer (SETR), which changes the previous segmentation model using a pure Transformer structure encoder instead of a CNN encoder. Inspired by the Vit model, SETR first performs the chunking process on the input image, and at the same time, it adds positional coding in the chunking process and converts the 2D encoder output vectors into 3D feature maps to achieve better segmentation results. Luo et al. (2022) proposed a network that utilizes mutual learning between CNNs and transformers for teaching and learning, which utilizes a “U” shaped network and the Swin transformer to collaborate in building a backbone model, a semi-supervised approach where the predictions of one network are chosen as pseudo-labels and then used to supervise the other network. Although the above studies have improved the segmentation effect to some extent, and some studies have combined CNN and Transformer, they cannot realize the fusion and dynamic interaction of multi-scale contextual information well, and the above networks have not entirely solved the problem of skin lesion segmentation.

Methods

Overview of our proposed architecture

Our proposed SUTrans-NET architecture is shown in Fig. 1, where our module consists of dual encoders to enhance the features in parallel and the CNN and Transformer branches in the dual encoders are dynamically fused at each layer, which breaks the limitations under the traditional dual encoder architecture and allows the network to take advantage of capturing both the global contextual information and the spatial information of the image. At the same time, the Zhang MHA module of the VIT architecture is replaced by the SGA module we developed to better capture the critical information of the image by adding spatial feature attention so that the network can focus more on feature-intensive tasks. The Squeeze-and-Excitation (SE) module is added to explicitly establish the correlation between channels and features to improve the segmentation quality. The network still adopts the traditional encoder–decoder architecture, and in the process of up-sampling, ASPP and slicing are added to obtain the multi-scale information to increase the sensory field of the image and to capture the more important information of the image to facilitate the segmentation of skin lesions.

Figure 1 The SUTrans-NET framework, which consists of a Transformer encoder on the far left, a CNN encoder in the middle, and a decoder on the far right, for extracting more discriminative local features and remote dependencies.

Dual encoder CNN with transformer

Medical images are characterized by small segmentation datasets, the need for sufficiently accurate segmentation results, fuzzy edges of lesions, etc. CNN networks can generate image representations that capture hierarchical patterns and obtain a global theoretical receptive field to distinguish the difference between foreground and background better. We use the pre-trained ResNet (He et al., 2016) as the backbone network of the CNN branch, the ResNet network mitigates the gradient vanishing problem in deep-level networks and speeds up the convergence speed when the network is trained, we use the ResNet50 network as part of the dual encoder, but the CNN network cannot capture more comprehensive contextual information since the CNN network relies on the effect very poorly at long distances. This severely limits the performance of the segmentation.

CNNs have limitations in obtaining long-range information in images, the attention mechanism in Transformer captures contextual information well, and Transformer’s variant networks, such as Segformer (Xie et al., 2021), SenFormer (Bousselham et al., 2021), and UCTransNet (Wang et al., 2022), have achieved good results in the field of segmentation. Inspired by these networks, we adopt VIT as the basic architecture of the encoder, and Fig. 2 shows the difference between our Transformer encoder and VIT. We keep the original overall architecture of Resnet, taking layer 0 as an example: the layer 0 features in the CNN are passed to the SpatialGroupAttention module of the Transformer layer 0 branches, and the Transformer merges the CNN features of the exact resolution and encodes them at the same time. In this way, the Transformer branch receives the results of the shallow texture information extracted by the CNN and passes them to the next layer of the Transformer branch structure. The CNN branch and the Transformer branch continuously exchange information globally and locally. Our network realizes the mutual interaction of the dual encoders, not only the encoding fusion at the last layer as in the previous network, which enables better dermatological segmentation.

Figure 2 The VIT encoder and our proposed encoder for SUTrans-NET, our model uses SpatialGroupAttention (SGA) module instead of Multi-Head Attention module, adding Squeeze Excitation (SE) module to focus on more important network information.

The idea of Focus downsampling is used instead of the patches idea of the VIT model.

Our basic network is influenced by the VIT model, the original VIT passes in an image of X ∈3×H×W, divides it into multiple patches of 16 ×16, divides each patch into fixed length vectors and feeds them into the Transformer, which transforms the problem of vision into a seq2seq problem, but inevitably makes the image lose its spatial features in the process of transformation However, the original patch embedding module turns the image into linear vectors and inevitably loses the spatial information of the image. We replace the patch embedding module in VIT with the Focus of yolov5 (Zhu et al., 2021), which slices the image and expands the number of channels by taking the value of the inter-pixel, which ensures the spatial characteristics of the image without changing the resolution of the original image, and at the same time, while being able to learn different sensory fields like CNN.

Ideas for SGA and SE module construction

We construct the SpatialGroupAttention (SGA) module in the network to replace the MHA module in the traditional VIT, which is shown in Fig. 3. In the traditional VIT, the feature vectors are susceptible to the influence of other similar feature vectors and noise leading to insufficiently fine segmentation results, and the attention focuses on some redundant information thus making the network parameters huge. The attention mechanism (Olshausen, Anderson & Van Essen, 1993) has demonstrated its importance in many tasks, including image localization (Bluche, 2016; Miech, Laptev & Sivic, 2017), sequence learning (Cao et al., 2015), etc. Computer vision aims to capture more critical representations of information attributes to build better spatial dependencies so spatial attention is incorporated into the network structure. ResNeXt (Xie et al., 2017) proposed the importance of grouping and that increasing the number of groups in a similar model can improve the accuracy value of the model. We use the idea of grouping attention integrating dimensional information to improve spatial attention, realize the semantics within the different group information learning, and improve the spatial distribution of features within groups. We get vectors similar to the target attention, while other unimportant positions become zero vectors.

Figure 3 SGA module grouping display.

We improve the grouping method based on the module (Li, Hu & Yang, 2019), SGE by putting all the information of the group directly into Batchsize, leaving only the features of a group to do the rest of the operations, such as some traditional operations for finding the mean, variance, sigmoid and other traditional operations. Our SGA module performs the same operation on each group so that each group learns the image features entirely and does not put all the information into Batchsize but performs strict feature grouping. In this way, the learning of sub-features within each grouping is improved to obtain the weight values of each grouping, and finally, the weight values are concatenated.

The position of the group in space is represented by a vector, x ={x1, …, m}, m = H × W, the mask is generated from the source of similarity between global and local features, and the global features are utilized to obtain the significant coefficients of the sub-features by means of dot product, which exhibits global and local similarity to some extent: (1) s=1m∑i=1mxi.

Getting the initial s for subtracting the mean divided by the standard deviation normalize and finally get the important weight values to get the important features in the picture.

Our network introduces the SE module (Hu, Shen & Sun, 2018), which focuses on the correlations and dependencies between channels and explicitly recalibrates the different feature channels. Through this mechanism, the SE module can utilize global information to highlight more noteworthy information while suppressing less critical image information, thus enabling the model to automatically learn important information about different channels. The SE module is divided into two parts, squeezing and excitation, where squeezing is based on the response of the channels to obtain a globally compressed feature volume, and all the layers use the information from the global sensory domain. Excitation, on the other hand, obtains the weight values for each channel through a simple gating mechanism and maps the weights to features as layer inputs for subsequent networks. In shallow networks, the SE module improves the quality of the lower layer information representation. In deeper networks, the SE can learn more features, flexibly recalibrate to different inputs, and accumulate by traversing the entire network to focus on more critical information features.

The SE module is essentially a computational unit, given the transformations constructed: Ftr:X → U, X ∈ RWt×Rt×Ct, U ∈ RW×R×C, Ftr denotes the standard convolution operator, the filter kernel is denoted by V = {v1, v2, …., vc}, the subscripts of v denote the filter parameters, and the output of Ftr is denoted by U = {u1, u2, …., uc}. (2) Uc=Vc∗X= ∑s=1cVcs∗Xs.

The ∗ symbol denotes the convolution, the sum of all channels produces the output, and the channel dependencies are embedded in Vc to recalibrate the filter response by establishing explicit channel dependencies. The main body of the SE module is divided into two steps : the Squeeze does the global information embedding, and the Excitation uses adaptive recalibration. Essentially, the global average pooling operation is used to perform global information embedding, to resolve the interaction dependencies of channels, to output signals from different channels to facilitate the operation of different filters on the local receptive fields, and to alleviate the problem of small receptive fields in low-level networks. The spatial information is compressed by global average pooling and the statistic z ∈ Rc is generated by reducing the spatial dimension of U by H ×W: (3) Zc=FsqUc=1H×W∑i=1H ∑i=1WUci,j.

The output of u is a collection of local descriptions, and we use such local descriptors to statistically inform and thus express the entire image, which was shown in Shen et al. (2015). It is a common phenomenon to use global average pooling for simple aggregation operations.

Excitation operation to fully capture channel dependence using gating mechanisms and sigmoid activation: (4) s=Fexz,W=σgz,W=σW2δW1z

(5) Xc ˜=FscaleUc,Sc=suuc.

where σ represents the Relu operation,W1∈Rcr×c, W2∈Rc×cr the parameter of the descending layer is set to W1, the parameter of the ascending layer is set to W2, and the scale factor is r, X ˜=X1 ˜,X2 ˜,…,XC ˜, Fscale(Uc, Sc) is the product of Uc ∈ RH×W and the corresponding channel of sc. The SE block interacts with the information by continuously stacking the convolutional layers to get the effect.

ASPP decoder

We introduced DeepLabV3 in the decoder Atrous Spatial Pyramid Pooling (ASPP) idea in the decoder to build Multi-scale Fusion Block (MSF Block), a large number of convolutional repetitive step-by-step and successive operations will lead to resolution reduction, and the uncertainty of localized image transform will hinder the dense prediction task, the MSF Block adopts parallel layout The MSF module adopts a parallel layout to extract features at different scales through multiple parallel null convolutional layers. The MSF module adopts a parallel layout to extract features at different scales by multiple parallel null convolutional layers, which are extracted at each individual branch to finally generate the results by fusion, different null rates to construct convolutional kernels with different sensory fields so that effective pixel point classification can be performed at each scale of the region, and different expansion factors to realize the extraction of multi-scale features so that multi-scale information can be effectively captured without too much loss of feature map resolution, and the resolution of the feature map can be reduced without too much loss of resolution. They are efficiently capturing multi-scale information without much loss of feature map resolution.

SUTrans-NET sets the void coefficient r to r (0, 3, 6, 9) for up-sampling, and slicing operation is used in the up-sampling process, and the CNN branch and the up-sampled branch are done two slicing operations each. Operations respectively to finally concat splice, because of its exact resolution can be perfect for different receptive fields for feature extraction, and we perform slicing processing on the channel, which can better integrate different We perform slicing on the channel to integrate features of different receptive fields better. Our decoder can better extract the image edge features than the traditional decoder.

Experiments and results

Datasets

The International Skin Imaging Collaboration (ISIC) collects publicly accessible dermoscopic image datasets. In this study, we verified the validity of SUTrans-NET by comparing experiments on ISIC 2016 (Gutman et al., 2016), ISIC 2017 (Codella et al., 2018) , and ISIC 2018 (Codella et al., 2019) public dermatological lesion datasets, and we also included the dataset PH2.ISIC 2016 had a total of 1,279 images, of which 900 were used for training images and 379 were used for testing. ISIC 2017 had a total of 2,750 images, of which 2,000 were used for training images and 600 were used for testing. The ISIC 2018 dataset used 2,594 RGB skin images for training images, and 1,000 were used for testing. The PH2 dataset included 200 8-bit RGB color skin mirror images, we randomly selected 160 images as the training set and the remaining 40 as the testing set.

We implemented SUTrans-NET on a single NVIDIA RTX 3090 GPU card. Our image input pre-network was reshaped to 224 × 224. For better initialization, we used Resnet 50 as the pre-trained model with an Adam optimizer, an initial learning rate of 0.0001, a batch size of 4, and a maximum epoch set to 100. To ensure the diversity of the samples, we use a data enhancement strategy that includes: horizontal, vertical, and row-column flipping, linear transformation, and random rotation. The network modules such as VIT, ASPP, and SE used in the experiments of this paper are open-source code provided by the corresponding author.

Evaluation metrics

In order to accurately evaluate the effectiveness of our model, we use the following evaluation metrics to evaluate the segmentation accuracy: ACCuracy (ACC), SEnsitivity (SE), SPecificity (SP), Intersection over Union (IoU) and Dice coefficient (Dice). (6) ACC=TP+FNTP+TN+FP+FN

(7) SE=TPTP+FN

(8) SP=TNTN+FP

(9) IoU=TPTP+FP+FN

(10) Dice=2⋅TP2⋅TP+FP+FN.

ACC is a prevalent evaluation metric, which is the number of correctly predicted samples divided by the number of all samples, with higher values representing better results. SE and SP’s value range is (0,1); the closer the value is to 1, the better the effect is. IoU is an evaluation metric for detecting the accuracy of the corresponding object; the value of IoU is the ratio of the lesion area to the segmentation result, the interval ∈ ∈ (0, 1); The closer you get to the 1 split, the better it works. Where TP and TN represent the number of lesion and background pixels correctly segmented, respectively, FN represents the value incorrectly predicted as a background pixel that was a skin lesion pixel at the time of prediction, and FP is the opposite.

Ablation studies

To demonstrate the effectiveness of different modules of our proposed SUTrans-NET network, we use ablation experiments to compare the enhancement effect of different modules, and our baseline model is a U-Net network using Resnet50 as one of the dual encoders, which solves the gradient vanishing problem well. We retain the structure of the traditional U-network and are also inspired by the period of the VIT network, and use a single NVIDIA RTX 3090 GPU card for the ablation experiments, strictly control the same experimental environment, and use the same data augmentation for the final network prediction to ensure the fairness of the experiments.

We conducted ablation experiments on the ISIC 2017 dataset, and the specific data are recorded in Table 1. The experiments were conducted in the following four main areas: (1) SUTrans_cnn represents the use of only CNN as an encoder to explore the segmentation effect of a single encoder.(2)SUTrans_vit uses only the modified VIT as a single encoder. (3) SUTrans_not_att explores the importance of our proposed SpatialGroupAttention (SGA) module on the segmentation results. (4) SUTran_not_aspp on the sampling effect of upsampling without ASPP. On the ISIC 2017 dataset, the IoU and Dice indices of the baseline model U-Net are 74.4% and 83.42%, respectively. In order to verify the effectiveness of our dual encoder, we test the performance of the network when only one of CNN encoder and Transformer encoder is included, in the network SUTrans_cnn when only CNN encoder is included, our method increases by 0.58% and Dice increases by 0.64% over the baseline model IoU, in the network SUTrans_vit when only Transformer encoder when the network SUTrans_vit, our method grows 5.52% and Dice grows 5.09% over its IoU, and we can see the effectiveness of the dual encoder for dermatologic segmentation. We remove the newly added SGA module (SUTrans_not_SGA), which has a 0.78% decrease in IoU and 0.61% decrease in Dice compared to our original network SUTrans-NET, which is still an improvement compared to the baseline model. We turn the ASPP module into a normal up-sampling module, which is 1.77% lower than the IoU and 1.58% lower than the Dice of our original network SUTrans-NET. The ablation experiments on the ISIC 2017 dataset show the improvement of our proposed dual encoder concerning the baseline model, and our newly proposed module SGA is also quite effective. The SUTrans-NET network plays a vital role in the segmentation performance, and our network is correct and effective.

Table 1 Statistical comparison of ablation experiments on key modules of the SUTrans-NET network.

Methods	Acc(%)	IoU(%)	Dice(%)	SE(%)	SP(%)	
U-Net	92.52	74.64	83.42	82.28	97.44	
SUTrans_cnn	94.09	77.78	85.68	84.46	96.32	
SUTrans_vit	92.21	72.86	81.23	79.81	97.97	
SUTrans_not_SGA	93.96	77.60	85.71	84.55	97.10	
SUTrans_not_aspp	93.35	76.61	84.74	82.30	97.81	
SUTrans-Net	93.90	78.28	86.32	86.18	97.56	
Notes.

Bold displays the values with the best performance in network comparison.

Table 2 Comparison of using different networks on ISIC- 2017 dataset.

Methods	Years	Acc(%)	IoU(%)	Dice(%)	SE(%)	SP(%)	
U-Net	2015	92.52	74.64	83.42	82.28	97.44	
DeLabV3+	2018	92.86	74.68	83.35	80.55	97.66	
EANet	2018	93.31	75.98	84.33	81.53	97.51	
AttUNet	2019	91.78	72.43	81.85	80.98	97.76	
FATNet	2021	93.44	76.17	85.39	83.92	94.93	
SwinUNet	2021	93.44	76.69	84.88	83.21	97.34	
TransUNet	2021	93.87	77.36	85.68	83.11	97.55	
SUTNet (ours)	2023	93.90	78.38	86.32	86.18	97.56	
Notes.

Bold displays the values with the best performance in network comparison.

Results of the ISIC-2017 dataset

On the ISIC-2017 dataset, we use the same experimental environment and data enhancement methods to compare seven more advanced semantic segmentation networks, including U-Net (Ronneberger, Fischer & Brox, 2015), DeepLabV3+ (Radman, Sallam & Suandi, 2022), EANet (Codella et al., 2019), AttU-Net (Oktay et al., 2018), FATNet (Wu et al., 2022), and SwinUNet (Chen et al., 2018), TransUNet (Chen et al., 2021b). Among them, U-Net, EANet, AttU-Net, and SwinUNet methods are newer medical image segmentation networks, and FATNet is specialized for skin lesion segmentation. In our comparison experiments, all contenders are run on the same computational environment and the same data extensions to ensure a fair comparison. EANet can achieve self-attention with a simple linear layer, AttU-Net proposes a novel Attention Gate (AG) model to automatically learn to focus on target structures of different shapes and sizes, and FATNet integrates an additional transformer branch to efficiently capture remote dependencies and global context information. Table 2 shows the comparative results of these state-of-the-art methods, which improve SE, SP, ACC, IoU, and Dice by 3.9%, 0.12%, 1.38%, 3.64%, and 2.90%, respectively, compared to the original baseline model U-Net,3.9%, 0.12%, 1.38%, 3.64% and 2.90%, respectively. Our method achieved good results on SE, SP, ACC, IoU and Dice evaluation metrics of 86.18%, 97.56%, 93.90%, 78.28% and 86.32%, respectively. Obviously, most of the indicators of our method are better than other methods and are highly competitive.

Meanwhile, we selected several typical networks to be visualized on the ISIC 2017 dataset, including U-Net, EANet, SwinUNet and TransUNet, several challenging networks, and the visualization results are shown in Fig. 4, which can be seen that: our method has a better segmentation effect at the edge of the lesion, and when the contrast between the lesion and the surrounding skin is low, the other four networks The segmentation effect is not apparent enough, and our segmentation area is closest to the lesion area. When the lesion area is prominent, the segmentation results of the four networks are more or less the same, but we are still competitive in the border area. Networks with a Transformer structure can extract global information well; our method combines the network advantages of CNN and Transformer so that the two branches form a dual encoder for dynamic interaction, and at the same time, it can capture the multi-scale information fusion so that the lesion edge segmentation becomes more apparent, and overall more robust, which proves the superior performance of SUTrans-NET in the superior performance in segmentation.

Figure 4 Comparison of ISIC-2017 segmentation results: (A) Input image, (B) ground truth, (C) U-Net, (D) EANet, (E) SwinUNet, (F) TransUNet, and (G) SUTrans-NET (Ours).

Where the red outline is the ground truth and the blue outline is the segmentation result of the specified method. Image source credit: ISIC 2017 dataset, https://challenge.isic-archive.com/data/#2017, CC0 (https://creativecommons.org/public-domain/cc0/).

Results of the ISIC-2016 dataset

On the ISIC-2016 dataset, we use the same experimental environment and data enhancement methods to compare five more advanced semantic segmentation networks, including U-Net (Ronneberger, Fischer & Brox, 2015), EANet (Guo et al., 2022), FATNet (Wu et al., 2022), SwinUNet (Cao et al., 2022), and TransUNet (Chen et al., 2021b). TransUNet is mainly used for medical image segmentation to recover local spatial information with the help of U-Net combination to realize precise localization. Table 3 shows the comparative results of these state-of-the-art methods, where the SUTrans-NET network improves SE, SP, ACC, IoU, and Dice by 1.92%, 0.05%, 0.48%, 1.52%, and 1.06%, respectively, compared to the original baseline model, U-Net. Our method achieved good results in SE, SP, ACC, IoU and Dice evaluation metrics with 93.66%, 96.19%, 95.90%, 85.48%, 91.43%, respectively. It is clear that our method obtained the highest scores in most of the indicators.

Table 3 Comparison of using different networks on ISIC- 2016 dataset.

Methods	Years	Acc(%)	IoU(%)	Dice(%)	SE(%)	SP(%)	
U-Net	2015	92.52	74.64	83.42	82.28	97.44	
EANet	2018	95.13	83.96	90.45	92.31	95.73	
FATNet	2021	95.48	84.69	91.04	90.98	96.23	
SwinUNet	2021	95.78	85.24	91.37	91.07	96.61	
TransUNet	2021	95.20	84.17	90.55	91.42	95.41	
SUTNet (ours)	2023	95.90	85.48	91.43	93.66	96.19	
Notes.

Bold displays the values with the best performance in network comparison.

We selected several typical networks to be visualized on the ISIC 2016 dataset, including U-Net, EANet, SwinUNet, TransUNet several challenging networks, and the visualization results are shown in Fig. 5. It can be seen that SwinUNet and TransUNet can reduce the effect of the weak ability to extract global contextual information, but for the challenging samples with complex boundaries may still produce false segmentation. For complex cases with different scales, irregular shapes, and fuzzy boundaries, our method still performs well on edges, and the dual encoder successfully captures both local features and global contextual information in extracting features, and SUTrans-NET still performs well on challenging images.

Figure 5 Comparison of ISIC-2016 segmentation results:(A) Input image, (B) ground truth, (C) U-Net, (D) EANet, (E) SwinUNet, (F) TransUNet, and (G) SUTrans-NET (Ours).

Where the red outline is the ground truth and the bule outline is the segmentation result of the specified method. Image source credit: ISIC 2016 dataset, https://challenge.isic-archive.com/data/#2016, CC0 (https://creativecommons.org/public-domain/cc0/).

Results of the ISIC-2018 dataset

On the ISIC-2018 dataset, we use the same experimental environment and data enhancement methods to compare five more advanced semantic segmentation networks, including U-Net, EANet, FATNet, SwinUNet, and TransUNet. Table 4 shows the results of the comparison of these advanced methods, compared to the original baseline model U-Net. SUTrans-NET network improves SE, SP, ACC, IoU, and Dice by 3.46%, 0.44%, 1.25%, 2.19%, and 2.24%, respectively. Our method achieved good results in SE, SP, ACC, IoU and Dice evaluation metrics, 91.93%, 94.02%, 93.33%, 81.63%, 88.58%, respectively. Obviously, most of the indicators of our method are better than other methods and have good competitiveness.

Table 4 Comparison of using different networks on ISIC- 2018 dataset.

Methods	Years	Acc(%)	IoU(%)	Dice(%)	SE(%)	SP(%)	
U-Net	2015	92.08	79.44	86.34	88.47	94.68	
EANet	2018	92.50	79.96	86.67	87.28	95.73	
FATNet	2021	92.86	80.90	87.74	91.48	93.19	
SwinUNet	2021	93.50	81.08	88.46	93.96	92.93	
TransUNet	2021	93.12	81.74	88.12	88.68	96.03	
SUTNet (ours)	2023	93.33	81.63	88.58	91.93	94.02	
Notes.

Bold displays the values with the best performance in network comparison.

In addition, we show several typical segmentation results obtained by different competitors by way of examples to facilitate visual comparison analysis. In our experimental study, we have selected four most representative visual comparison methods, including U-Net, FATNet, EANet, and SwinUNet for several challenging networks, which are demonstrated in Fig. 6. It is worth observing that our method usually outperforms other competitors in these demonstrations, especially when challenging examples are involved, our method is able to achieve the best segmentation results. Even for skin lesions of different scales and irregular shapes, our method still achieves the best segmentation results, very close to the ground truth.

Figure 6 Comparison of ISIC-2018 segmentation results: (A) Input image, (B) ground truth, (C) U-Net, (D) EANet, (E) SwinUNet, (F) TransUNet, and (G) SUTrans-NET (Ours).

Where the red outline is the ground truth and the blue outline is the segmentation result of the specified method. Image source credit: ISIC 2018 dataset, https://challenge.isic-archive.com/data/#2018, CC0 (https://creativecommons.org/public-domain/cc0/).

Results of the PH2 dataset

Finally, we performed further comparative experiments on the PH2 dataset to validate the robustness of our proposed SUTrans-NET. Unlike the three aforementioned datasets, the PH2 dataset here contains only a few hundred images from the dermatology department of the Pedro Hispano Hospital in the Matosinhos region of Portugal. To ensure the fairness of this smaller dataset, we did comparative experiments on five networks, UNet R2UNet, BiONet (Xiang et al., 2020), UNeXt (Valanarasu & Patel, 2022), and SegNetr (Cheng et al., 2023), in the same environment. R2UNet uses recurrent convolutional networks and recurrent residual convolutional networks to improve the segmentation accuracy, BiONet reuses the constructed modules in a recurrent way without increasing the parameters, UNeXt uses a labeled MLP block to label and project the convolutional features and to model them, and SegNetr can dynamically perform local global interactions at any stage, and all the above are the different methods for medical image segmentation. The statistical comparison results of the different methods on the PH2 dataset are shown in Table 5. Compared with the original baseline model U-Net, the SUTrans-NET network improves SE, IoU, and Dice by 1.9%, 2.3%, 1.34%, respectively. Our method achieved good results in SE, SP, ACC, IoU and Dice evaluation metrics, 86.71%, 91.77%, 94.59%, 87.39%, and 92.56%, respectively. It is obvious that most of the indicators of our method are better than other methods and have good competitiveness. To guarantee a fair comparison on this relatively small dataset, we implemented all competitors under the same computing environments and with the same data augmentations in our comparison experiments.

Table 5 Comparison of using different networks on PH2 dataset.

Methods	Years	Acc(%)	IoU(%)	Dice(%)	SE(%)	SP(%)	
U-Net	2015	94.90	85.09	91.22	94.72	96.03	
R2UNet	2018	94.84	83.34	90.37	93.23	96.56	
BiONet	2020	91.83	78.32	86.69	90.29	94.75	
UNeXt	2022	94.24	85.03	91.29	96.65	91.17	
SegNetr	2023	94.24	85.38	91.42	96.74	92.04	
SUTNet (ours)	2023	94.59	87.39	92.56	96.71	91.77	
Notes.

Bold displays the values with the best performance in network comparison.

The visualization results of the different methods on the PH2 dataset are shown in Fig. 7. We compare four methods, UNet, BiONet, UNeXt, and SegNetr. The UNet method is significantly less effective in segmenting the inconspicuous regions, and BiONe introduces bi-directional jump connections, which helps the encoder to process semantic features in the decoder. UNeXt improves the segmentation results by means of a tokenized MLP block to label and project convolutional features, and uses MLP to model the representation, improving the segmentation effectiveness. SegNetr is a newer and lighter network with 59% and 76% fewer parameters and GFLOPs, respectively, compared to regular U-Net. However, the above methods still cannot effectively capture and utilize the global context information, which hinders the further improvement of accuracy. Our method obtains better segmentation performance by sharing local features and remote dependencies through global and local dual encoders, which still provides good segmentation results in the case of blurred boundaries.

Figure 7 Comparison of PH2 segmentation results: (A) Input image, (B) ground truth, (C) U-Net, (D) BiONet, (E) UNeXt, (F) SegNetr, and (G) SUTrans-NET (Ours).

Where the red outline is the ground truth and the blue outline is the segmentation result of the specified method. Image source credit: PH2 dataset, https://challenge.isic-archive.com/data/#2018, CC0 (https://creativecommons.org/public-domain/cc0/).

Discussion

The contrast between skin lesions and normal skin is often low, and the irregular size and shape makes skin lesion segmentation a tough challenge. Several scholars have proposed various methods in recent years. Most traditional methods use a single encoder to extract feature information, and only CNN or Transformer architectures are used to complete the encoder construction, which still can not extract the local and global context information well. SUTrans-NET constructs interactive CNN and Transformer dual encoders. CNN can generate image representations that capture the hierarchical patterns and obtain the global theoretical acceptance area. Transformer excels in capturing long-range dependencies in feature maps.

From the combined results of the experiments on the four datasets, the U-Net model based on the CNN network is slightly less effective than the network based on Transformer architecture when segmentation is performed, which indicates the importance of the attention mechanism for segmentation. We construct the SGA module instead of the MHA module of the Transformer architecture, which has a noticeable effect enhancement in the case of the lower number of participants, and the SGA module utilizes the idea of grouping attention, which makes the groups able to integrate more noteworthy information to do spatial attention enhancement. The introduced SE module enables better integration between the local context and global context and also suggests that researchers can return their attention to the overall Transformer architecture; the Transformer lacking the MHA module still has much potential, and our results show that the method of constructing the spatial information can help the performance enhancement of VIT, which will be a new direction, Tang & Yu (2021) also provides new ideas for VIT research.

On the other hand, our network has some limitations in some edges where the skin lesions are not apparent in the background, and the network still cannot distinguish the natural tissue from the lesion tissue with certainty when the contrast between the two is low. However, our results are still significantly better than the other network segmentation and the closest to the ground truth.

Conclusions

In this article, we propose a novel encoder–decoder network, SUTrans-NET, designed to address the challenging task of skin lesion segmentation. Specifically, SUTrans-NET constructs a dual encoder to perform image segmentation. Unlike existing dual decoders, our CNN and Transformer branches constantly interact dynamically to realize the sharing of both local and global information, and the parallel dual decoder efficiently extracts the shallow texture features and global context information, which significantly increases the sensory field. We use the designed SGA module instead of the MHA module to save memory while grouping the attention and extracting the vital information of the image, improving the learning ability of sub-features, and enhancing the ability of multi-scale context modeling. We conducted extensive experiments on four public datasets (ISIC 2016, ISIC 2017, ISIC 2018 and PH2 datasets) to evaluate our proposed skin lesion segmentation method, and the experimental results proved that SUTrans-NET achieved some success and better accuracy performance in the skin lesion segmentation task, and the experimental results proved that SUTrans-NET achieved some success and better accuracy performance in the skin lesion segmentation task. We will focus on the overall architecture of Transformer rather than just improving the performance of self-attention, which may be a new direction, and we will continue to explore more possibilities of SUTrans-NET in skin lesion segmentation and apply it to more areas.

Supplemental Information

Supplemental Information 1 Source code

Additional Information and Declarations

Competing Interests

Author Contributions

Data Availability

The authors declare there are no competing interests.

Yaqin Li conceived and designed the experiments, performed the experiments, performed the computation work, prepared figures and/or tables, authored or reviewed drafts of the article, and approved the final draft.

Tonghe Tian conceived and designed the experiments, performed the experiments, performed the computation work, prepared figures and/or tables, and approved the final draft.

Jing Hu analyzed the data, prepared figures and/or tables, authored or reviewed drafts of the article, and approved the final draft.

Cao Yuan conceived and designed the experiments, analyzed the data, prepared figures and/or tables, authored or reviewed drafts of the article, and approved the final draft.

The following information was supplied regarding data availability:

The code is available in the Supplemental File and the data is available at Zenodo: Tiantonghe. (2023). SEUForemer_code. Zenodo. https://doi.org/10.5281/zenodo.8320546.

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
