# Peer review of "SUTrans-NET: a hybrid transformer approach to skin lesion segmentation"

_PeerJ Computer Science, doi:10.7717/peerj-cs.1935_

## Round 0.1 · original submission · Major Revisions

Dear authors,

After a meticulous review of the SUTrans-NET: A hybrid transformer approach to skin lesion segmentation publication proposal, the decision is to proceed to Major Revisions.

The decision is based on the reviews provided that meet some of my own concerns, especially in the design and validity of the findings. I suggest that you address all these points in order to be reflected in the next version to be sent.

Kind regards,
PCoelho

**Language Note:** The review process has identified that the English language must be improved. PeerJ can provide language editing services - please contact us at [email protected] for pricing (be sure to provide your manuscript number and title). Alternatively, you should make your own arrangements to improve the language quality and provide details in your response letter. – PeerJ Staff

Reviewer 1 ·

Basic reporting

The paper proposes a new skin lesion segmentation network, SUTrans-NET, which combines CNN and Transformer in a parallel fashion to form a dual encoder, where both CNN and Transformer branches perform dynamic interactive fusion of image information in each layer. The paper is interesting, and the introduction, methodology and results are interesting, but the manuscript must be proofread. However, the discussion must be more exhaustive, and the results must be compared with the literature.

Experimental design

no comment

Validity of the findings

no comment

Cite this review as

Reviewer 2 ·

Basic reporting

This paper is poorly written and requires a major improvement in terms of its writing.
1. Missing spaces between sentences. Examples: line 28, 34, 35, 112, 121, 160, 172, 234, 364 etc
2. Grammer errors: A few examples:
- line 156: "U-Net(Ronneberger et al., 2015) It has been"
- line 211: "by it. Network for supervision. "
- line 335: "a pretraining model"
3. Inconsistent citation style: For example, in line 172, it starts with "Almasni et al.", but in line 175, the sentence started with "(Wang et al., 2020)".
4. Formatting and inconsistency of mathematical definitions
- line 281 x={x1,...,m} and Equation (1) -> Please use subscripts instead of xi.
- Inconsistency of mathematical definitions: In the paragraph between line 295-296, it was first assumed that 'U \in R^{W×R×C}'. But in line 300, it says ' U_c \in R^{H×W}'. In addition, in Equation (4), W is used as a weight matrix.
- Intersection over Union was denoted as 'IoU' in line 344 and 345, but as 'Iou' in line 367,369, 370.

Experimental design

The paper is about a new network for skin lesion segmentation, called SUTrans-NET, which combines convolutional neural networks (CNN) and Transformer in a dual encoder structure. The paper claims that this network can capture both local and global information of the images, and improve the segmentation accuracy and robustness. The paper also introduces a multi-grouping module SpatialGroupAttention to enhance the Transformer branch. The paper evaluates the proposed network on four datasets: ISIC 2016, ISIC 2017, ISIC 2018 and PH2, and shows that it outperforms existing methods.

Validity of the findings

The authors compared their approach SUTrans-NET with various existing methods across multiple datasets. Despite the tasks being fundamentally similar, they chose different existing methods to compare with for different datasets. The specific methods that the authors chose for comparison on each dataset need to be clarified. Furthermore, the authors did not specify the training strategies employed for these existing methods, and whether these strategies were consistent with their own method. The following are the methods that the authors compared:

ISIC- 2016 dataset: U-Net, EANet, FATNet, SwinUNet, TransUNet.
ISIC 2017 dataset: U-Net, DeLabV3+, EANet, AttUNet, FATNet, SwinUNet, TransUNet.
ISIC- 2018 dataset: U-Net, EANet, FATNet, SwinUNet, TransUNet.
PH2 dataset: UNet, R2UNet, BiONet, UNeXt, and SegNetr.

Cite this review as

---

## Round 0.2 · Minor Revisions

Dear authors,

You are advised to critically respond to all comments point by point when preparing a new version of the manuscript and while preparing for the rebuttal letter. Please address all the comments/suggestions provided by the reviewers.

Kind regards,
PCoelho

Reviewer 1 ·

Basic reporting

The authors considered the previous comments, and the manuscript can be accepted.

Experimental design

no comment

Validity of the findings

no comment

Cite this review as

Reviewer 3 ·

Basic reporting

The work presents a hybrid structure that combines the advantages of CNN and Transformer for segmenting images related to skin cancer. In my opinion, the work is interesting and very relevant to the medical field. It has a good bibliographical background, is well written in English and well structured. The introduction is good and the figures are important.

Experimental design

no comment

Validity of the findings

The results are relevant and the validation process is robust, since the method is applied to different databases and compared with other approaches.

Additional comments

I have few specific comments for consideration and correction by the authors:

1. There are some problems with the placement of commas and spacing. Examples in lines: 4, 33, 42, 52, 80, 87, 113, 120, 298, 305, 329, 410, 446. I suggest that the authors review the text in detail to eliminate these typos.

2. In the Introduction, the text on lines 49 and 50 refers to some traditional methods. It would be appropriate to name some of these methods and cite references that prove their ineffectiveness.

3. The bibliographical reference on lines 513 and 514 (Abbas, et al. (2011a)) is written exactly the same as the following reference on lines 515 and 516.

Cite this review as

---

## Round 0.3 · Minor Revisions

Dear authors,

Please address all the comments/suggestions provided by the reviewers.
The improvement/revision of the writing is advised.

Kind regards,
PCoelho

**Language Note:** The Academic Editor has identified that the English language must be improved. PeerJ can provide language editing services - please contact us at [email protected] for pricing (be sure to provide your manuscript number and title). Alternatively, you should make your own arrangements to improve the language quality and provide details in your response letter. – PeerJ Staff

Reviewer 1 ·

Basic reporting

The authors considered the previous comments, and the manuscript can be accepted.

Experimental design

N/A

Validity of the findings

N/A

Additional comments

N/W

Cite this review as

Reviewer 3 ·

Basic reporting

The authors have well completed items 2 and 3 of my suggestions for revision, however there are still some typos in relation to item 1. Therefore, before publication I suggest that they make the following corrections in relation to the spacing of citations:

Line 41: Replace "problem,(Ashour et al., 2018)proposed" with "problem, (Ashour et al., 2018) proposed a neutral..."

Line 50: Replace "methods, (Dar and Padha, 2019)they" with "...methods, (Dar and Padha, 2019) they..."

Line 54: Replace "... (HRNet)(Sun et al., 2019)" with "(HRNet) (Sun et al., 2019)..."

Line 56: Replace "(Radman et al., 2022)proposed" with "(Radman et al., 2022) proposed".

Line 89: Replace "U-Transformer(Petit et al., 2021)" with "U-Transformer (Petit et al., 2021)"

Line 122: Replace "(ASPP)(Chen et al., 2017)" with "(ASPP) (Chen et al., 2017)"

Line 159: Replace "U-Net(Ronneberger et al., 2015)" with U-Net (Ronneberger et al., 2015)"

Line 163: Replace "segmentation.(Ma et al., 2021)" with "segmentation. (Ma et al., 2021)"

Line 168 : Replace "(Soulami et al., 2021)proposed" with "(Soulami et al., 2021) proposed"

Line 174: Replace "MCNet(Wang et al., 2020)," with "MCNet (Wang et al., 2020),"

Line 175: Replace "UNet.(Al-Masni and Kim, 2021)" with "UNet. (Al-Masni and Kim, 2021)"

Line 201: Replace "semantics.(Tang et al., 2021)" with "semantics. (Tang et al., 2021)"

Line 236: Replace "ResNet(He et al., 2016)" with "ResNet (He et al., 2016)"

Line 244: Replace "SenFormer(Bousselham et al., 2021)," with "SenFormer (Bousselham et al., 2021),"

Line 300: Replace "in(Shen et al., 2015)." with "in (Shen et al., 2015)."

Line 331: Replace "2016(Gutman et al., 2016)," with "2016 (Gutman et al., 2016),"

Line 331: Replace "(Codella et al., 2019)public" with "(Codella et al., 2019) public"

Line 411: Replace "U-Net(Ronneberger et al., 2015)," with "U-Net (Ronneberger et al., 2015),"

Line 412: Replace "EANet(Guo et al., 2022)," with "EANet (Guo et al., 2022),"

Line 412: Replace "SwinUNet(Cao et al., 2022)" with "SwinUNet (Cao et al., 2022)"

Experimental design

'no comment'

Validity of the findings

'no comment'

Additional comments

'no comment'

Cite this review as

---

## Round 0.4 · accepted · Accept

Dear authors, we are pleased to verify that you meet the reviewer's valuable feedback to improve your research.

Thank you for considering PeerJ Computer Science and submitting your work.